# Detection of SARS-CoV-2 Derived Small RNAs and Changes in Circulating Small RNAs Associated with COVID-19

**DOI:** 10.3390/v13081593

**Published:** 2021-08-11

**Authors:** Claudius Grehl, Christoph Schultheiß, Katrin Hoffmann, Mascha Binder, Thomas Altmann, Ivo Grosse, Markus Kuhlmann

**Affiliations:** 1Institute of Computer Science, Martin-Luther-University Halle-Wittenberg, 06120 Halle, Germany; claudius.grehl@informatik.uni-halle.de (C.G.); ivo.grosse@informatik.uni-halle.de (I.G.); 2Department of Internal Medicine IV-Oncology/Hematology, Martin-Luther-University Halle-Wittenberg, 06120 Halle, Germany; christoph.schultheiss@uk-halle.de (C.S.); Mascha.Binder@uk-halle.de (M.B.); 3Institute of Human Genetics, Martin-Luther-University Halle-Wittenberg, 06120 Halle, Germany; katrin.hoffmann@uk-halle.de; 4Department of Molecular Genetics, Leibniz Institute of Plant Genetics and Crop Plant Research (IPK), Heterosis, OT Gatersleben, 06466 Stadt Seeland, Germany; altmann@ipk-gatersleben.de

**Keywords:** SARS-CoV-2, COVID-19, long COVID, small RNA, circulating-miRNA

## Abstract

Cleavage of double-stranded RNA is described as an evolutionary conserved host defense mechanism against viral infection. Small RNAs are the product and triggers of post transcriptional gene silencing events. Up until now, the relevance of this mechanism for SARS-CoV-2-directed immune responses remains elusive. Herein, we used high throughput sequencing to profile the plasma of active and convalescent COVID-19 patients for the presence of small circulating RNAs. The existence of SARS-CoV-2 derived small RNAs in plasma samples of mild and severe COVID-19 cases is described. Clusters of high siRNA abundance were discovered, homologous to the nsp2 3′-end and nsp4 virus sequence. Four virus-derived small RNA sequences have the size of human miRNAs, and a target search revealed candidate genes associated with ageusia and long COVID symptoms. These virus-derived small RNAs were detectable also after recovery from the disease. The additional analysis of circulating human miRNAs revealed differentially abundant miRNAs, discriminating mild from severe cases. A total of 29 miRNAs were reduced or absent in severe cases. Several of these are associated with JAK-STAT response and cytokine storm.

## 1. Introduction

A novel disease has been spreading in a pandemic manner since the end of 2019. This disease was named COVID-19 (coronavirus disease 2019), as it was found to be associated with the zoonotic Severe Acute Respiratory Syndrome Corona Virus type 2 (SARS-CoV-2). SARS-CoV-2 is an enveloped, positive-sense, single-stranded RNA betacoronavirus of the family *Coronaviridae*. In humans, coronaviruses cause infections of the respiratory tract and can lead to acute respiratory distress syndrome (ARDS), which may result in widespread inflammation of many organs and death [1].

The case fatality ratio of SARS-CoV-2 is approximately 100 times higher than the common influenza infection (case fatality ratio: 0.01–0.04 deaths per 100 positive cases, [2]). It ranges between 1 and 5 per 100 cases, depending on local medicinal standards (MERS: 34.4, SARS; 9.5 [3]) and vulnerability, e.g., Trisomy 21 [4]. This high rate is mainly driven by the lack of effective treatment of severe cases.

Cleavage of double-stranded RNA is a well-studied host mechanism, involved in viral defense [5,6] in the plant and animal kingdoms. This initial defense reaction is triggered by the presence of double-stranded RNA (dsRNA). Presence of dsRNA is a typical indication for virus infection, occurring by back-folding or hybridization of RNA plus and minus strands. dsRNA can be cleaved by specific enzymes (DICERS) and generate small RNAs of a specific size. The existence of small RNAs with potential miRNA function has been described for various viral genomes [7]. For the SARS coronavirus, an agonistic function of *nsp15* against the detection of the dsRNA intermediates has been described [8]. These short interfering RNAs (siRNA) are products of virus inactivation, but also trigger silencing and degradation of homologous virus sequences via the RNA-induced silencing complex (RISC) [9].

The involvement of small RNAs in SARS defense, and their ability to inhibit replication has been described [10,11] and also proposed for SARS-CoV-2 [12]. One miRNA-like small RNA was reported to be encoded in the nsp3 region of the SARS-CoV-2 genome [13]. Until now, no detailed analysis of the presence, abundance and persistence of SARS-CoV-2 derived small RNAs in infected tissue of patients with COVID-19 has been reported.

Small RNA from plasma of control and SARS-CoV-2 positive patients was analyzed. Beside the potential virus-derived small RNAs, circulating human microRNAs (circ-miRNAs) were present in the investigated samples. Several of these miRNAs are described as potent biomarkers for diseases [14,15]. High throughput small RNA sequencing of plasma was applied, reads were mapped against the virus genome and miRNA signatures, and differential abundant small RNAs were evaluated (Appendix A). Plasma was the tissue of choice because: (I) prior publications indicated the presence of circulating small RNAs [12,16,17], (II) substantial DICER expression was described in cells of the lung and blood (Appendix A) and (III) SARS-CoV-2 infection causes multi-organ effects, with symptoms in the lung, heart, kidney, muscles and neural system that is mirrored in plasma composition [18].

Here, we are addressing the questions:Do we see circulating small RNAs corresponding to parts of viral genome, and can they be used as marker for ongoing or cleared infections?Do we see circulating small RNAs corresponding to parts of viral genome with distinct features?Do we see circ-miRNAs that suggest RNA-based regulation/fine-tuning of the host immune response that differs in mild and severe cases, with potential as biomarker [14,15]?

## 2. Methods

### 2.1. Patients and Samples

Blood was collected during March and August 2020 at the University Hospital Halle (Saale) from patients (Caucasian) in close proximity to Halle (Saale) in Germany.

Plasma was isolated by centrifugation of whole blood from EDTA vacutainers for 15 min at 2000× *g*. Samples were stored at −80 °C before use. Blood collection was performed under institutional review board approvals, numbers 2020-039 and 11/17, in accordance with the Declaration of Helsinki. Informed written consent was obtained from all participating patients or legal representatives. In total, 100 µL of plasma were used for the heat inactivated analysis, according to [19] and processed by GenXPro (Frankfurt am Main, Germany).

### 2.2. Detection of SARS-CoV-2 Antibody in Human Plasma

Levels of antibodies directed against the spike (S1) and the nucleocapsid protein (NCP) of SARS-CoV-2 in plasma were determined using the Anti-SARS-CoV-2-ELISA IgA/IgG and Anti-SARS-CoV-2-NCP-ELISA ELISA kits from Euroimmun AG (Lübeck, Germany), according to manufacturer’s instructions.

### 2.3. RNA Isolation, Library Prep and Sequencing

Plasma RNA was extracted using an internal purification protocol of GenXPro GmbH, based on silica columns. The small RNA libraries for Illumina sequencing were prepared using the TrueQuant small RNA kit (GenXPro GmbH, Frankfurt am Main, Germany) as described by [20]. The libraries were sequenced using an illumina NextSeq500 instrument (GenXPro GmbH, Frankfurt am Main, Germany). For detection of SARS-CoV-2 associated small RNAs, the raw reads of the individual samples, as well as the grouped samples (mild/severe) were mapped against the SARS-CoV-2 genome sequence (NCBI Genome Reference Sequence NC_045512) in forward and reverse orientation. CLC-Workbench 11 software was used with default settings, allowing one mismatch for mapping of reads against the virus genome.

## 3. Results

To screen for SARS-CoV-2-associated small RNAs, plasma was collected from eight PCR-positive COVID-19 patients within the first three weeks after symptom onset, as well as from one healthy donor and one patient with bacterial pneumonia as controls. Five of the COVID-19 patients had severe courses and required extracorporeal membrane oxygenation (ECMO) in an intensive care unit (ICU). Three of the five severe cases succumbed to the disease. The remaining three had mild to moderate disease symptoms. Of these, three were sampled during active disease phase within the first 17 days after symptom onset, and one in the convalescence phase 41 days post infection. Two samples during active infection were available for one of the mild cases, with a sample each in the active and in the convalescence phase for one patient. The sampling, clinical and laboratory findings of the cohort and NGS read output are summarized in Table 1, Appendix A, Tab Samples and Appendix A. Comprehensive immunological data are described for five of these patients [21].

### 3.1. Small SARS-CoV-2 Derived RNAs Are Detectable in Plasma of COVID-19 Patients

For the identification of small SARS-CoV-2 derived RNAs, we analyzed plasma of patients with PCR-confirmed presence of SARS-CoV-2 (mild and recovered, *n* = 5; severe, *n* = 5) and negative controls (*n* = 2) without a present or previous SARS-CoV-2 infection. RNA was extracted, the fraction of small RNAs enriched and subjected to library preparation. The read number after adapter trimming ranged from 0.8 to 6.9 million (Table 1) with a median of 1.4 million reads per sample.

For detection of SARS-CoV-2 associated small RNAs, these raw reads of the individual samples, as well as the grouped samples (severe/mild), were mapped against the SARS-CoV-2 genome sequence in sense and antisense orientation, allowing one mismatch (Appendix A for numbers and sequences of mapped reads in sense and antisense).

The summarized mapping results are displayed in Table 2. In mild/recovered and severe samples substantial small virus-derived RNAs are detected. The results show *n* = 85 (sense: 68, antisense: 17) in mild cases and *n* = 171 (sense: 118, antisense: 53) in severe cases in both orientations, while the sense orientation is higher in abundance. The size of the mapped RNAs ranged from 15 to 19 nts. The healthy individual did not reveal small RNAs complementary to the SARS-CoV-2 genome. The pneumonia disease control had three reads mapping to less SARS-CoV-2 specific regions. One of these reads was assigned to the polyA structure in the 3′-region of the virus genome, and the second was mapped to the 3′-end of the SARS-CoV-2 genome which is homologous to e.g., Vibrio and Acinetobacter genomes. Acinetobacter baumannii is a known cause of both community- and hospital-acquired infections, including pneumonia [22]. The 15 nt long antisense read shares complete homology to the human ERCC excision repair 6 like-transcript, and might represent a degradation product.

The read counts were normalized to RPM (reads per million) for quantification. The abundance of reads per sample is displayed in Figure 1. The highest number of reads (35 RPM) was detected in the sample of a mild case 10 days after first symptoms (pt68), where the lowest number of read output was also noted. In addition, a high abundance of SARS-CoV-2 reads was found in the severe cases in pt1 and pt44 (5 and 74 days after infection, ~10 RPM). In samples of recovered patients (pt7_4 and pt50), virus-associated RNAs were detectable (27 and 41 days after infection, respectively). A low abundance of reads was found across the virus sequence. However, particular clusters of reads were detectable in the 5′-region of the virus genome associated with the *nsp2* encoding sequence.

It is interesting to note that in all patients where the SARS-CoV-2 specific antibodies were not [yet] present (Appendix A), the abundance of virus dependent small RNAs was higher (pt1, pt44 and pt68, Appendix A).

### 3.2. Mapping of Small Virus-Derived RNAs Reveal a Cluster in the 3′-Region Coding for nsp2 and Four 19 nt Long Small RNA Sequences

Two particular clusters of reads are found to be associated with the SARS-CoV-2 genome sequence. Both are located in the *nsp2* encoding region (Figure 2A), in a forward orientation within ORF1. Reads associated with cluster 1 (CCTTCAATGGGGGATGTCC) were detected in all PCR positive tested samples. A higher abundance of cluster 2 reads (ACCCTAAGAGGGGTGTAT), located upstream of cluster 1, was only detected in severe cases. The normalized abundance of cluster 1 matching reads is shown in Appendix A. Furthermore, the highest number of reads was found in the sample of a mild case (pt68), 10 days after infection. No reads of this cluster were found in the pneumonia patient and the healthy individual. In samples pt7_4 and pt50, representing recovered cases, reads from this cluster were detectable. A detailed graphical description of cluster 1 and 2 are given in Appendix A.

As human microRNAs can have a size between 19 and 24 nts [23], we were focusing on the 19 nt long small RNA associated with SARS-CoV-2. Here, four sequences were identified, two of them with a higher abundance. Relative positions in the virus genome are displayed in Appendix A.

The first sequence with a high abundance was CCTTCAATGGGGGATGTCC (at position 1058 in the virus genome, 10 reads), which was also located in cluster 1. A sequence specific target gene search by (psRNATarget [24]), adjusted for the human transcriptome, revealed six genes with an expectation score < 4 (Appendix A). Among these six genes, *GFRA1* (GDNF family receptor alpha-1; MIM 601496) was found, which has been reported as a receptor required for taste perception [25] and function of the olfactory system [26].

Using the sequence CATATTCAGTGGATGGTTG (pos. 9616, 11 reads) of the higher abundant 19 nucleotide-sized RNA as input for the target search, we found *ARL6IP6*, *TRPM3*, *HTR2A* and *FGF2* among the seven top potential targets (Appendix A). *ARL6IP6* (MIM 616495) is an ADP-Ribosylation-Like Factor 6 Interacting Protein 6, associated with ischemic stroke [27]. *HTR2A* (MIM 18135) is a Serotonin 5-HT-2A Receptor with effects on sleep quality [28]. Reduction of *TRPM3* (MIM 608961) is described to be associated with chronic fatigue syndrome/Myalgic encephalomyelitis patients [29], and *FGF2* (MIM 134920) has been proposed to play a role in intussusceptive angiogenesis during COVID-19 [30].

The third sequence with low abundance was CTCATTCAAGGAGGACTTA (pos. 24945), with five potential target genes predicted (Appendix A).

The fourth sequence was CAATGCTCATGGATTGTTG (pos. 29496) in antisense orientation at the 5′-end of ORF9 encoding the nucleocapsid. For this sequence, eight potential target RNAs were identified (Appendix A). Among these, *PSG1* and *SLFN13* were found: *PSG1* (MIM 176390) is a pregnancy-specific beta-1-glycoprotein 1, regulating *P1GF* and *VREGF-A*. The upregulation of both genes has been described as a COVID-19 marker [31]. *SLFN13* (MIM 614957) is a member of the SCHLAFEN family, and is suggested to mediate antiviral responses to both influenza A and B RNA virus infections [32].

### 3.3. Analysis of Circulating Human miRNA Showed Differences in Samples of Mild and Severe Cases

Circulating small RNAs in the blood are described as potent biomarkers for several diseases and general inflammatory reactions [12,14,15,16,17].

For a detailed analysis of circulating miRNAs in the sample set, the reads were mapped against the publicly available miRBase [33], revealing information for long non-coding RNAs, transfer-RNAs, structural RNAs and miRNAs.

Principal component analysis (PCA) indicated a common separation of all SARS-CoV-2 infection samples from the healthy individual (HD, Appendix A). The sample from the pneumonia patient (ICU2, negative control) is among the group of SARS-CoV-2 infection samples. As the patient was affected by a bacterial infection, this might indicate a common pro-inflammatory pattern of circ-miRNA. The patients which succumbed to the disease (pt1, pt25 and pt44) have the most diverse circ-miRNA pattern, and the samples are located apart from the core cluster.

Differentially abundant circ-miRNAs were identified by comparison of the set of mild cases (*n* = 3) with severe cases (*n* = 5, *p* < 0.05, DESeq [34], Appendix A). Twenty of the thirty-one differentially abundant miRNAs showed a significant (*p* < 0.05) lower abundance (log2fold change) for severe cases. Eleven circ-miRNAs had positive log2fold-change values, indicating higher abundance in severe cases. In total, we found 1119 Homo sapiens mircroRNAs (hsa-miRNAs), 17,871 human Ensembl transcript ids (ncRNAs) and 392 tRNAs. According to our analysis, 210 ensemble transcripts (including a high number of non-coding RNAs) were differentially abundant, while only four tRNAs (tRNA-His-GTG-1-4, tRNA-Lys-TTT, tRNA-Arg-TCT and tRNA-Ser-GCT: ~3 log2fold decreased) showed a significant differential abundance (*p* < 0.05). The complete list of differentially abundant RNAs is given in Appendix A.

Differentially abundant circ-miRNAs sorted by log-fold change and association are shown in Table 3. The available literature reference for their identified context is given in Appendix A.

The diagrams in Figure 3 describe the detailed read abundance for the top six differentially abundant circ-miRNAs: has-miRNA320a-3hashsa-miRNA629-5p, hsa-miRNA29a-3p, has-miRNA-324-3p, has-miRNA-185-5hasnd hsa-miRNA4516, whereby the first five of the six clearly distinguish mild from severe cases in our cohort. hsa-miRNA320 (Figure 3) was reported to be unregulated, in association with heart failure in patients with diabetes mellitus [35,36,37]. The entire subset of miRNA320 (b,c,d) was found to be up-regulated in severe cases here.

hsa-miRNA29a-3p was reported as a biomarker for the diagnosis of tuberculosis [38].

has-miRNA625-3p is used as biomarker for Malignant Pleural Mesothelioma [39] (up-regulated) and Nontuberculous Mycobacterial Pulmonary Disease [40] (up-regulated). Its counterparhashsa-miRNA625-5p, was reported to suppress inflammatory responses by targeting *AKT2* (MIM 164731) in human bronchial epithelial cells [41]. This miRNA was found to be strongly down-regulated (−1.64 log2fold change, *p* = 0.0468) in all severe cases.

hsa-miRNA4516 was reported to be involved in *STAT3* (MIM 102582) regulation and also associated with psoriasis [42].

hsa-miRNA19a (Figure 3) was absent in the severe samples analyzed. hsa-miR19 is involved in many steps of the inflammatory response and coupled to cytokine response [43]. Due to its absence in severe samples and its low read number, hsa-miR19 did not show up among the DESeq generated table of significant DE miRNAs.

In addition, hsa-miR-122-3p and hsa-miR-122b-5p were found among the differential abundant circ-miRs (−4 log2fold change, *p* = 0.02), which are already proposed as a potential target of human miRNAs in SARS-CoV-2 for RNA-based drug discovery [44].

## 4. Discussion

### 4.1. SARS-CoV-2 Derived Small RNAs Are Detectable

The mapping of the plasma-derived small RNAs reads against the SARS-CoV-2 genome resulted in the identification of SARS-CoV-2 derived sequences. These sequences are uniquely present in samples of SARS-CoV-2 positive patients, and are considered as SARS-CoV-2 derived small RNAs. The size of the identified reads ranged from 15 to 19 nts (Appendix A). These reads are relatively short in length compared to the expected size of 21 +/− 1 nts [45,46] for DICER products. Up to now, the length of 21 +/− 1 nts has been considered a hallmark of DICER products. This difference in size might be an explanation of why other studies failed to observe differences in virus-derived small RNAs [47]. However, based on their size, we were not able to identify the mechanistic origin of these molecules in the cell-free plasma. One option might be that partial degradation, cleavage or misprocessing is generating artificial byproducts of virus replication. Another origin might be processing or generation of smaller products by DICER and miRNA processing, followed by an export process. This is supported by the presence of human miRNAs (ranging from 16 to 27 nts) with a different size than the canonical 21mers miRNAs as products of the human DICER [23] in the serum. Some of the identified sequences are overlapping with in silico predicted regions encoding small RNAs [48], based on folding and cleavage mechanisms. As shown in Appendix A, in all our tested patients we saw a negative relation between the presence of SARS-CoV-2 specific antibodies and the abundance of SARS-CoV-2 derived small RNAs. In the samples (pt1, pt67-1 and pt68) taken at an early timepoint after infection, a substantial/high abundance of small RNAs was detectable. Although the number of analyzed sample was low, the presence of virus-derived small RNAs suggests dicing of viral dsRNA as an early defense mechanism (4–10 days after infection) for virus suppression. Cleavage of dsRNA might represent a first hurdle, inactivating the intruding viruses, while the antibody defense is not yet established.

The identified virus-derived sequences of 19 nts length have the potential to mimic human miRNAs and suppress target mRNA function, based on their complementarity. The encoding of miRNA-like small RNAs in a viral genome [7] was described for the bovine leukemia virus [49], the Ebola virus [50] and lately for SARS-CoV-2 [13]. Among the predicted target mRNAs, *GFRA1* (GDNF family receptor alpha-1), *HTR2A*, *TRPM3* and *FGF2* are found. Post transcriptional gene suppression mediated by the virus-derived small RNAs of these transcripts might explain some of the COVID-19 associated symptoms. These transient symptoms, such as loss of taste and smell, insomnia, chronic fatigue syndrome and elevated risk for thrombosis [51] might be associated with the transient suppression of the identified transcripts. This finding opens a novel perspective on the molecular regulation and importance of small virus-derived RNAs as a regulator for virus related processes in the host. In contrast to [13], no virus-derived small RNAs larger than 20 were detected in our serum samples.

The comparison of the SARS-CoV-2 mapped reads with the human genome validated the virus specificity. Only the low-complexity reads with mappings to the unstructured polyA virus region showed an additional mapping to human sequences. The presence of these sequences in the samples of control patients without a SARS-CoV-2 infection is also indicative of their origin from the human genome, and therefore an artificial random alignment to the virus genome. Therefore, these reads are considered misaligned and are excluded from the SARS-CoV-2 derived small RNAs.

Considering the ongoing evolution of the SARS-CoV-2, reflected by its entropy and mutation rate [52], it is interesting to note that a high level of entropy (0.5) is detectable at position 1058–1060 in close proximity to the identified small RNA cluster. However, up to now, the mutations of novel detected variants have not affected the region covered by the small RNA cluster.

Due to the small number of samples, we were only able to show the presence of the virus-derived small RNAs, and it was not possible to show a quantitative correlation with patient gender or duration of infection.

### 4.2. Small Virus-Derived RNAs from the Cluster Might Be Usable as Biomarkers

A cluster of reads with a higher abundance was found homologous to the 3′-end of the *nsp2* encoding region.

These SARS-CoV-2 specific reads might be suitable as biomarkers for an early, ongoing and cleared infection. The detection of a virus-derived microRNA as a biomarker for severe COVID-19 cases was described [13]. The size and abundance of the clustered small RNAs are relatively high, and comparable to some human miRNAs. As such, they can be detected and quantified by stem-loop qPCR [53]. As identified in samples pt7_4 and pt50, they are substantially detectable in patients that have recovered from COVID-19 21- and 48-days post-infection. In these previously SARS-CoV-2 PCR positive patients, no RNA virus particles were detectable with standard PCR methods anymore when the samples for the siRNA analyses were taken. Although the recurrence of RNAemia is still under debate [54,55,56], it is generally accepted that viral RNAs can be detectable, independent of COVID-19, up to three weeks after infection [57]. Therefore, the presence of the detected virus-derived small RNAs is in in agreement with reported RNA presence. The abundance of the small RNAs from the cluster region (Figure 2B) is comparable to other miRNAs, proposed as serum-derived biomarkers [58], e.g., hsa-miR7-5p, hsa-miR370-3p and hsa-miR221-5p.

### 4.3. Severe Cases of COVID-19 Are Associated with Differential Abundance of Specific Circulating Human miRNAs

MicroRNAs are small RNA molecules with a suppressive function on their target mRNAs. Their expression is regulated by developmental as well as external factors. Their extracellular presence in serum has been described previously, although the detailed function of the most circulating miRNAs is still elusive. However, their presence and abundance are used as biomarkers, associated with disease responses [12,14,15,16,17,59,60]. Here we analyzed the abundance of the circ-miRNAs and compared mild versus severe COVD-19 cases in cell free plasma. As shown in Table 3 and Appendix A (modified after [43]), the amount of several circ-miRNAs associated with inflammation responses are changed in severe COVID-19 cases in contrast to the mild ones. This refers to mir-4516, miR-320 and miR-145b-5p, for example. This strong response at the miRNA level contributing to the immune response regulation agrees with the clinical description of patients requiring ICU admission [61,62]. This involves higher concentrations of GCSF, IP10, MCP1, MIP1A and TNFα, suggesting that the cytokine storm was associated with disease severity. Based on an in silico analysis [63], miR362-5p (3.6 log2fold change, *p* = 0.009, Table 2) was proposed for targeting the RNA encoding the virus receptor ACE2 and let7F2, targeting the RNA encoding trans-membrane-protein TMPRSS2. Both proteins were found to be involved in the SARS-CoV-2 infection processes. While miR362-5p abundance is increased in ICU cases, let-7F is reduced (Appendix A). No overlap of differential abundant miRNAs from blood tissue is found [64]. 

Furthermore, several symptoms of COVID-19 might be correlated with the differential abundance of some circulating miRNAS, as the subfamily of hsa-miRNA320 was found to be increased in severe cases (Figure 3). The complete miRNA320 family is used as biomarker for lung-associated diseases [65], and is down-regulated upon adenovirus infection [66]. A low abundance of hsa-miRNA320b was described as a predictive biomarker for survival rate of COPD patients [67]. Therefore, the SARS-CoV-2 induced deregulation of the miRNA320 family is involved in the infection response, reflecting the molecular status of the lung tissue.

hsa-miRNA625-3p (Figure 3), a circ-miRNA used as biomarker for Malignant Pleural Mesothelioma [39] (up-regulated) and Nontuberculous Mycobacterial Pulmonary Disease [40] (up-regulated) was found to be strongly down-regulated (−1.64 log2fold change, *p* = 0.0468) in all severe cases. Its counterpart, hsa-miRNA625-5p, was reported to suppress the inflammatory responses by targeting AKT2 in human bronchial epithelial cells [41].

Overexpression of hsa-miR-502-3p (here: 1.7 log2fold) was described to be involved in the regulation of the STAT3 pathway [68].

hsa-miR-363-3p (here: −2.5 log2fold) was described as a biomarker for an acute coronary syndrome [69]. In addition, miRNA hsa-miR-126-3p was found to be reduced in severe cases. This finding was recently confirmed by qPCR analysis, comparing healthy persons and COVID-19 patients [70]. hsa-miR-126-3p (here: −2.1 log2fold) has been shown to control the DNA damage response by repressing ATM protein kinase activity in endothelial cells, and is also a possible contributor to endothelial and lung tissue dysfunction [71].

Our data give a first insight into a novel field of SARS-CoV-2 virus defense. We are aware that the cohort size was very limited. Within our data sample set, we are dealing with a sex bias and have only limited information about characteristics and other predispositions that may affect the levels of circulating microRNAs; therefore, the presented results should be seen as a first insight, and further conclusions might require additional experimental pursuits to prove.

## Figures and Tables

**Figure 1 viruses-13-01593-f001:**
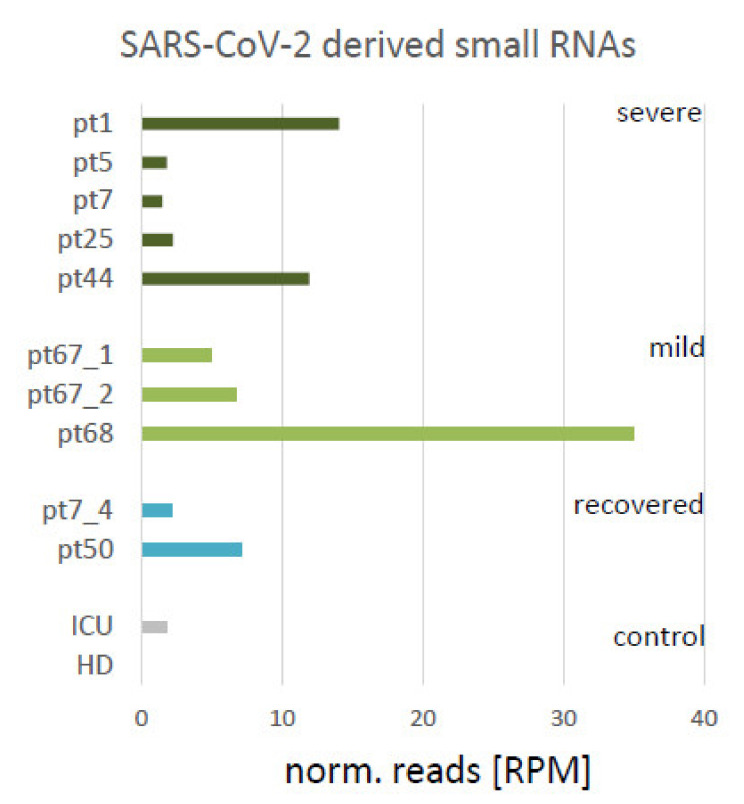
SARS-CoV-2 derived small RNAs. Bar diagram indicating the normalized number of reads from the samples in reads per million (RPM) mapping to SARS-CoV-2 genome sequence, and therefore considered as virus-derived. Active/severe COVID-19 (ICU): dark green, mild COVID-19: light green, recovered from COVID-19: turquoise, control samples (negative for presence of SARS-CoV-2 by PCR-test): grey bars.

**Figure 2 viruses-13-01593-f002:**
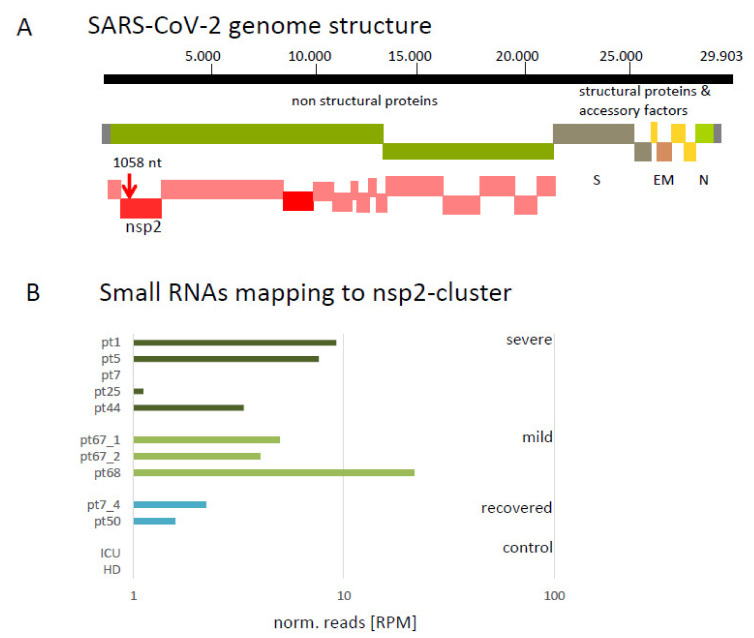
Cluster of SARS-CoV-2 derived small RNAs. (**A**) Schematic representation of SARS-CoV-2 genome. The red vertical arrow indicates the identified cluster 1. The cluster 1 is located at position 1058 nt from the genome start in sense orientation at the 3′ end of the sequence coding for the non-structural protein 2 (*nsp2*, red bar) within open reading frame 1 (ORF1, green bar). *nsp*: non-structural protein; ORF: open reading frame; Structural proteins: S: spike; E: envelope; M: membrane; N: nucleocapsid. (**B**) Bar diagram indicating the normalized number of reads from the samples in reads per million (RPM) mapping at the cluster region (sequence: CCTTCAATGGGGGATGTCC). Control samples (negative for presence of SARS-CoV-2 by PCR-test, blue bars; mild COVID-19, light grey; severe COVID-19 (ICU), dark grey.

**Figure 3 viruses-13-01593-f003:**
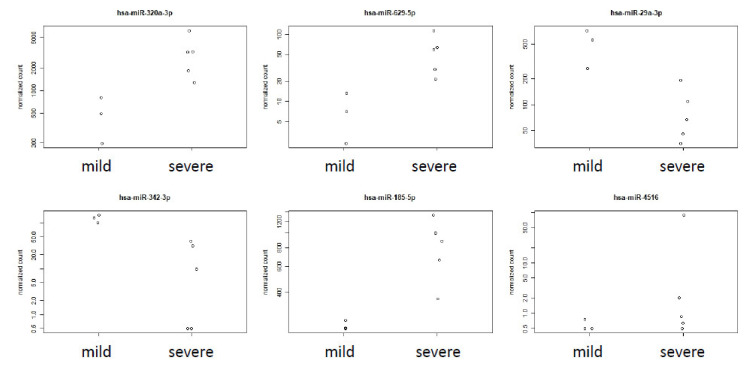
Abundance of hsa-miR-320a-3p, hsa-miR-629-5p, hsa-miR-29a-3p, hsa-miR-342-3p, hsa-miR185-5p and hsa-miR-4516 in plasma of mild and severe COVID-19 patients. Dot plot diagrams indicating the normalized number of reads in [RPM] (reads per million), hsa-miR-320a-3p, hsa-miR-629-5p, hsa-miR-29a-3p, hsa-miR-342-3p, hsa-miR185-5p and hsa-miR-4516. Samples were extracted from the list of differentially abundant miRNAs (*p* < 0.05), comparing mild versus severe samples.

**Table 1 viruses-13-01593-t001:** Overview of samples and generated reads. Pt: patient (yellow: male; turquoise: female), pos: positive, neg: negative, dai: presumed day after infection. Color-code: severe symptoms at the timepoint of sampling, dark green; mild symptoms at the timepoint of sampling, light green; recovered after symptoms, light blue; control, grey.

ID	Gender	COVID-19	Sampling (dai)	SARS-CoV2 PCR	COVID-19 Symptoms	Color-Code	Reads
pt1	male	pos.	5	pos.	severe, dead		1849274
pt5	male	pos.	20	pos.	severe		2230128
pt7	female	pos.	15	pos.	severe		1360989
pt25	male	pos.	16	pos.	severe, dead		2694258
pt44	male	pos.	74	pos.	severe, dead		6872831
pt67_1	female	pos.	3	pos.	mild		3442766
pt67_2	female	pos.	17	pos.	mild 2nd sample		1498621
pt68	female	pos.	10	pos.	mild		829230
pt7_4	female	pos.	27	neg.	recovered 2nd sample		1364983
pt50	female	pos.	41	neg.	recovered after mild		1273895
ICU	female	neg.	0	neg.	none		1132230
HD	female	neg.	0	neg.	none		1318929

**Table 2 viruses-13-01593-t002:** Mapping results of plasma small RNA reads mapping to SARS-CoV-2 genome. SARS-CoV-2 genome describes the number of small RNA sequences and unique reads annotated to the SARS-CoV-2 genome in sense (top) and antisense (bottom) orientation. Dark green: severe cases, light green: mild cases, grey: control.

	NGS	SARS-CoV-2 Genome			
		Sense		Antisense	
Category	Reads	Reads	Sequences	Reads	Sequences
severe	13.65 × 10^7^	118	40	53	23
mild	8.41 × 10^7^	68	16	17	8
control	2.45 × 10^7^	2	2	1	1

**Table 3 viruses-13-01593-t003:** List of hsa-miR with a significant difference in abundance between mild and severe COVID-19 cases. Green: upregulated, red: downregulated in severe cases.

	log2FoldChange	*p*-Value
hsa-miR-4516	5.338	0.00493
hsa-miR-362-5p	4.099	0.00912
hsa-miR-548k	3.726	0.02429
hsa-miR-320a-3p	2.672	0.00033
hsa-miR-320b	2.484	0.00517
hsa-miR-320c	2.339	0.01413
hsa-miR-320d	2.167	0.04336
hsa-miR-185-5p	1.812	0.00484
hsa-miR-629-5p	3.072	0.00067
hsa-miR-1180-3p	2.521	0.02330
hsa-miR-502-3p	1.710	0.03886
hsa-miR-454-3p	−3.235	0.01749
hsa-miR-625-3p	−1.879	0.04092
hsa-miR-30b-5p	−1.893	0.03071
hsa-miR-192-5p	−1.934	0.03201
hsa-miR-451a	−1.945	0.01957
hsa-miR-197-3p	−1.955	0.01970
hsa-miR-29b-3p	−2.080	0.04121
hsa-miR-126-3p	−2.114	0.01316
hsa-miR-146b-5p	−2.118	0.01188
hsa-miR-30c-5p	−2.226	0.01768
hsa-miR-144-5p	−2.444	0.00713
hsa-miR-29a-3p	−2.481	0.00100
hsa-miR-363-3p	−2.508	0.01054
hsa-miR-99a-5p	−2.523	0.00570
hsa-miR-342-3p	−2.753	0.00373
hsa-miR-193b-3p	−2.828	0.03472
hsa-miR-190a-5p	−3.118	0.03146
hsa-miR-365b-3p	−3.523	0.02085
hsa-miR-122b-5p	−3.960	0.02326
hsa-miR-122-3p	−4.128	0.02050

## Data Availability

The original reads sequences are available for download at: https://doi.ipk-gatersleben.de/DOI/42ac1f14-ab8e-440f-bd7b-0faa465cfbce/39046ce0-3b02-4428-9083-f92360d0b2cb/2/1847940088. DOI of the dataset released in this manuscript were constructed using the e!DAL system [72].

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
