# Peer review of "Detection of SARS-CoV-2 Derived Small RNAs and Changes in Circulating Small RNAs Associated with COVID-19"

_viruses, 2021, doi:10.3390/v13081593_

Round 1
Reviewer 1 Report
Interesting study and highly relevant to the ongoing SARS-CoV2 pandemic in which authors used RNAseq method to profile plasma of active and convalescent COVID-19 patients for the presence of small circulating RNAs. Identification of circulating microRNAs potentially could be used as biomarkers to differentiate between patients that would progress towards development of severe or mild COVID-19 disease. However, the cohort of samples used in the study is not ideal to answer stated questions due to very small number of samples, gender bias, no cohort characteristics. Also, there is high variability in the days after infection ranging from 3 to 74 days post-infection without information whether patients already developed severe vs mild symptoms. One suggestion to make stronger conclusion is to validate some of the identified microRNAs by qRT-PCR method.
Major comments:
- Very small cohort of patients with different stages of SARS-CoV2
- No demographics of the cohort with basic characteristics and other predispositions that may have affected the levels of circulating microRNAs, esp. host microRNAs
- No validation of the identified circulating microRNAs by qRT-PCR to support the findings
- Total number of NGS reads (Table 2) substantially different between “severe”, “mild” and “control” in the decreasing order. Similarly, sequences mapping to SARS-CoV-2 genome decrease in the same order which could be attributed to the differences in the total number of reads
Minor comments:
- RNA isolation, library prep and sequencing needs to be written in its own section in Methods and Materials as it is currently under “Detection of SARS-CoV-2 antibody in human plasma”
- All tables need to be properly formatted: removed red squiggly lines; define abbreviations (eg. Sampling DAI in Table 1); capital letters
- Table 1 – blue/grey group has not been defined that consists of two samples from “recovered patients”
Author Response
Dear Katarina Dragić, dear reviewers
I would like to re-submit a revised version of the manuscript “Detection of SARS-CoV-2 derived small RNAs and changes in circulating small RNAs associated with COVID-19
for consideration of publication in Viruses,
by authors
Claudius Grehl, Christoph Schultheiß, Katrin Hoffmann, Mascha Binder, Thomas Altmann, Ivo Grosse & Markus Kuhlmann
On behalf of the authors, I have to say that we are grateful for your encouraging assessment of our initial manuscript and would like to especially thank the anonymous reviewers for their critical evaluation and very helpful suggestions for improvement. We have carefully edited our manuscript according to the recommendations made by the reviewers and hope that it is now suitable for final acceptance by Viruses.
Please find below a listing of the comments made by the reviewers and descriptions of how we dealt with them when revising the manuscript.
Reviewer 1
Open Review
(x) I would not like to sign my review report
( ) I would like to sign my review report
English language and style
( ) Extensive editing of English language and style required
(x) Moderate English changes required
( ) English language and style are fine/minor spell check required
( ) I don't feel qualified to judge about the English language and style
|
Yes |
Can be improved |
Must be improved |
Not applicable |
|
|
Does the introduction provide sufficient background and include all relevant references? |
(x) |
( ) |
( ) |
( ) |
|
Is the research design appropriate? |
( ) |
(x) |
( ) |
( ) |
|
Are the methods adequately described? |
( ) |
(x) |
( ) |
( ) |
|
Are the results clearly presented? |
( ) |
(x) |
( ) |
( ) |
|
Are the conclusions supported by the results? |
( ) |
( ) |
(x) |
( ) |
Comments and Suggestions for Authors
Interesting study and highly relevant to the ongoing SARS-CoV2 pandemic in which authors used RNAseq method to profile plasma of active and convalescent COVID-19 patients for the presence of small circulating RNAs. Identification of circulating microRNAs potentially could be used as biomarkers to differentiate between patients that would progress towards development of severe or mild COVID-19 disease. However, the cohort of samples used in the study is not ideal to answer stated questions due to very small number of samples, gender bias, no cohort characteristics. Also, there is high variability in the days after infection ranging from 3 to 74 days post-infection without information whether patients already developed severe vs mild symptoms. One suggestion to make stronger conclusion is to validate some of the identified microRNAs by qRT-PCR method.
- We are thankful to this positive and encouraging view on our work.
Major comments:
- Very small cohort of patients with different stages of SARS-CoV2
- We agree in the comments made by the reviewer. He indeed points to the weakness of our dataset. We are also aware of the point that our cohort of patients is relatively small, but however sufficient for the description of presence of virus derived small RNAs. In order to make the reader aware of this weakness in our study we addressed these weaknesses in the end of the discussion:
- No demographics of the cohort with basic characteristics and other predispositions that may have affected the levels of circulating microRNAs, esp. host microRNAs
- So far possible, we added the requested information:
“Blood was collected during March and August 2020 at the University Hospital Halle (Saale) from patients (Caucasian) in close proximity to Halle (Saale) in Germany.” (line105)
and stated in the discussion
“Our data give first insight into a novel field of SARS-CoV-2 virus defense. We are aware that the cohort size is very limited. Within our data sample set we are dealing with a sex bias and have only limited information about characteristics and other predispositions that may affect the levels of circulating microRNAs, therefore the presented results should be seen as first insight and further conclusions might require additional experimental prove.“ line 375ff
More detailed Information is given in Table S1 Tab. Samples (linked in line 139)
- No validation of the identified circulating microRNAs by qRT-PCR to support the findings
- We agree that validation would be helpful and also additional investigation of the detected virus derived small RNAs after recovery will be interesting, but within the given time limit new experiments will not be possible.
- Total number of NGS reads (Table 2) substantially different between “severe”, “mild” and “control” in the decreasing order. Similarly, sequences mapping to SARS-CoV-2 genome decrease in the same order which could be attributed to the differences in the total number of reads
- As different numbers of samples were subjected to the analysis of severe and mild cases we were only focusing here on the qualitative presence of these reads. For the depiction of SARS-CoV-2 derived reads in individual samples we used Figure 1 and Figure 2, showing normalized read counts. But also here, due to the limited number of samples no correlation is detectable and we agree that quantitative statements are difficult.
The Sentence: “Due to the small number of samples we are only able to show the presence of the virus derived small RNAs, but no quantitative correlation with the patient gender or duration of infection is possible.” (line 318) was added.
Minor comments:
- RNA isolation, library prep and sequencing needs to be written in its own section in Methods and Materials as it is currently under “Detection of SARS-CoV-2 antibody in human plasma”
- Section moved and added.
- All tables need to be properly formatted: removed red squiggly lines; define abbreviations (eg. Sampling DAI in Table 1); capital letters
- Table 1 – blue/grey group has not been defined that consists of two samples from “recovered patients”
- We edited all tables according reviewer1 comments (automated correction was switched off) and added the requested information in the legends.
We again thank all reviewers for the constructive criticism.
We hope that we could address all raised points and improve the manuscript for final acceptance in Viruses.
Yours Sincerely,
Markus Kuhlmann
Reviewer 2 Report
The paper is well done and appropriately describes the analytical procedures, I would like you to describe the subdivision according to the patients degree of disease into severe, mild and recovered.
The methods paragraph for a better reading of the paper I prefer to follow the introduction paragraph.
Author Response
Dear Katarina Dragić, dear reviewers
I would like to re-submit a revised version of the manuscript “Detection of SARS-CoV-2 derived small RNAs and changes in circulating small RNAs associated with COVID-19
for consideration of publication in Viruses,
by authors
Claudius Grehl, Christoph Schultheiß, Katrin Hoffmann, Mascha Binder, Thomas Altmann, Ivo Grosse & Markus Kuhlmann
On behalf of the authors, I have to say that we are grateful for your encouraging assessment of our initial manuscript and would like to especially thank the anonymous reviewers for their critical evaluation and very helpful suggestions for improvement. We have carefully edited our manuscript according to the recommendations made by the reviewers and hope that it is now suitable for final acceptance by Viruses.
Please find below a listing of the comments made by the reviewers and descriptions of how we dealt with them when revising the manuscript.
Reviewer 2
Open Review
( ) I would not like to sign my review report
(x) I would like to sign my review report
English language and style
( ) Extensive editing of English language and style required
( ) Moderate English changes required
(x) English language and style are fine/minor spell check required
( ) I don't feel qualified to judge about the English language and style
|
Yes |
Can be improved |
Must be improved |
Not applicable |
|
|
Does the introduction provide sufficient background and include all relevant references? |
(x) |
( ) |
( ) |
( ) |
|
Is the research design appropriate? |
(x) |
( ) |
( ) |
( ) |
|
Are the methods adequately described? |
( ) |
(x) |
( ) |
( ) |
|
Are the results clearly presented? |
(x) |
( ) |
( ) |
( ) |
|
Are the conclusions supported by the results? |
(x) |
( ) |
( ) |
( ) |
Comments and Suggestions for Authors
The paper is well done and appropriately describes the analytical procedures, I would like you to describe the subdivision according to the patients degree of disease into severe, mild and recovered.
- We are thankful to this positive and encouraging view on our work.
The methods paragraph for a better reading of the paper I prefer to follow the introduction paragraph.
- We moved the methods paragraph to the requested place.
We again thank all reviewers for the constructive criticism.
We hope that we could address all raised points and improve the manuscript for final acceptance in Viruses.
Yours Sincerely,
Markus Kuhlmann
Round 2
Reviewer 1 Report
I am satisfied with authors' revisions